# Actin-Associated Proteins and Small Molecules Targeting the Actin Cytoskeleton

**DOI:** 10.3390/ijms23042118

**Published:** 2022-02-14

**Authors:** Jing Gao, Fumihiko Nakamura

**Affiliations:** School of Pharmaceutical Science and Technology, Tianjin University, Tianjin 300072, China; gaojinggj@tju.edu.cn

**Keywords:** actin, cytoskeleton, actin-binding protein, actin-associated proteins

## Abstract

Actin-associated proteins (AAPs) act on monomeric globular actin (G-actin) and polymerized filamentous actin (F-actin) to regulate their dynamics and architectures which ultimately control cell movement, shape change, division; organelle localization and trafficking. Actin-binding proteins (ABPs) are a subset of AAPs. Since actin was discovered as a myosin-activating protein (hence named actin) in 1942, the protein has also been found to be expressed in non-muscle cells, and numerous AAPs continue to be discovered. This review article lists all of the AAPs discovered so far while also allowing readers to sort the list based on the names, sizes, functions, related human diseases, and the dates of discovery. The list also contains links to the UniProt and Protein Atlas databases for accessing further, related details such as protein structures, associated proteins, subcellular localization, the expression levels in cells and tissues, mutations, and pathology. Because the actin cytoskeleton is involved in many pathological processes such as tumorigenesis, invasion, and developmental diseases, small molecules that target actin and AAPs which hold potential to treat these diseases are also listed.

## 1. Introduction

Since actin was first discovered as a myosin-activating protein, actin itself has been recognized in a sense as an actin-binding protein (ABP), because actin can polymerize to form filamentous actin (F-actin). Monomeric globular actin (G-actin) is an ATPase, where the binding of ATP and the hydrolysis of ATP to ADP regulates the assembly and disassembly of F-actin. ABPs regulate not only these processes but also the organization of F-actin in bundling, cross-linking, branching, anchoring, and scaffolding. Recent reviews described the discovery, structure, and mechanism of assembly and disassembly of actin in detail at a molecular level [1,2,3]. Structures and functions of ABPs have recently been reviewed as well [1,2]. Not all AAPs are ABPs. For example, transmembrane proteins, scaffolding proteins, and enzymes that post-translationally modify actin are not usually considered to be ABPs, even though they directly interact with actin. Here however, we have listed all the actin-associated proteins (AAPs) whose direct interactions with actin have been biochemically demonstrated. Such a sortable list of AAPs including their names, sizes, functions, related human diseases, and dates of discovery should be useful in the comprehensive understanding of AAPs. The list also contains links to the UniProt and Protein Atlas databases for accessing more details related to AAPs such as their structures, associated proteins, subcellular localization, expression levels in cells and tissues, mutations, and pathology. Readers may download the list (Word and Excel formats are provided) to sort and group the information according to their needs. This review also lists the small molecules that interact with actin and AAPs to regulate their functions, thereby holding potential for use as research reagents and clinical drugs.

## 2. Discovery of AAPs

Although a nearly complete human genome sequence is available [4], protein and DNA sequence-based computational analyses are not amenable to identifying many AAPs. For example, synaptopodin 2-like protein is a member of the synaptopodin family and highly co-localizes with actin filaments but does not directly interact with actin, unlike other members of the family [5]. Therefore, laborious biochemical works remain the primary means for identifying AAPs, as evidenced by the slow increase in the number of AAPs discovered each year (Figure 1).

Appendix A summarizes the known AAPs with their protein names, gene names, locations on the chromosome, numbers of amino acid residues, binding modes, functions, related human diseases, actin-binding domains, links to UniProt and the Human Protein Atlas, years found, and references. For some AAPs, the functions, binding modes, and actin-binding domains have not been determined and remain blank in the table; the table is also provided as an Excel file (Appendix A). When multiple names are used for a single AAP because of independent discoveries or discoveries in different species, the protein and gene names registered in UniProt are indicated together with other known names. If splice variant(s) are known, the number of amino acid residues of the canonical isoform is indicated. The binding modes are categorized by commonly used functional terms, namely monomer binding, nucleation, polymerization, depolymerization, scaffolding, stabilization, severing, annealing, capping, cross-linking, branching, anchoring, and motor (Figure 2). Some of the members of the AAP protein family have lost the ability to interact with actin and are not listed in the table. For example, the myosin heavy chain 16 (MYH16) in humans lost muscle contractile ability, and it is not clear if it still interacts with actin, whereas MYH16 in non-human primates is functional for having powerful jaw muscles [6]. Many ABPs use calponin-homology (CH) domain(s) to interact with F-actin but also use unique domains, making it difficult to computationally predict the actin-binding domain (ABD) in the human genome sequence. The UniProt and Human Protein Atlas databases possess protein sequences, structures, subcellular localizations, interacting proteins, mutations, pathologies, and more. The first report of each AAP and review manuscript, if any, are cited in the reference section.

Myosin, now a well-known ABP, was described in 1864 as a muscle protein and an enzyme that has ATPase activity but not as an ABP, because actin had not been discovered yet at that time [3]. Rather, actin was isolated as an activating protein (hence named actin) for the ATPase activity of myosin in 1942, which identifies myosin as the first ABP [3]. Since G- and F-actin were recognized in 1945, actin itself was identified as the second ABP [7]. Myosin found in non-muscle human blood platelets closely resembles the corresponding muscle myosin [8]. More ABPs, such as alpha-actinin and troponin, were isolated from skeletal muscle, but the third non-muscle ABP was isolated from rabbit alveolar macrophages and named “actin-binding protein.” It is now known as filamin A [9]. The authors did not expect at that time that non-muscle cells would express so many ABPs. Since then, 403 AAPs that include all family members encoded on different genes have been reported. They are listed in Appendix A.

## 3. Functions of AAP

Actin filaments are right-handed helices that consist of two strands of globular molecules. The polarity is visualized by a myosin S1 fragment, which creates barbed (+) and pointed (−) ends. Actin polymerization can be initiated from both preexisting F-actin and G-actin. In preexisting F-actin, elongation occurs faster at the barbed end than at the pointed end. When polymerization starts from G-actin, the polymerization does not occur in a linear fashion but in a curve with an initial slow lag phase. The lag phase is attributed to an actin nucleation reaction that consists of a dimer formation and a following trimer formation. Once nucleated, growth phase continues until a steady state is reached [2].

The functions of ABPs are often categorized by how they interact with actin and regulate its dynamics [2,10]. In this review, we have grouped them as shown in Figure 2 and indicated in Appendix A.

The nucleating factors, such as the formin family of proteins, nucleate unbranched actin filaments by stabilizing actin dimers, whereas Arp2/3 complex facilitates the nucleation of branched filaments by mimicking dimer formation. Human protein spire homolog 1/2 that was originally identified in Drosophila [11] and leiomodin promote nucleation by bringing multiple actin monomers [12]. The Ena/VASP family proteins enhance actin nucleation using tandem WH2 (WASP homology 2) motifs that bind to actin [13]. However, high concentrations of actin (50–500 μM) in the cytosol should be spontaneously polymerized without a nucleating factor. Inhibition of such polymerization is achieved by the use of monomer binding proteins, such as thymosin-β4 and profilin [14,15,16,17], and capping proteins, such as gelsolin [18] and tropomodulin [19], to maintain a monomer concentration at ~25–100 μM. The monomer binding protein can sequester actin monomers, preventing them from the polymerization of F-actin. The concentrations of thymosin-β4 (300–600 μM) and profilin (10–50 μM) with high affinity to G-actin are high enough to perform this task [20,21,22]. The binding of the capping protein to the growing barbed end of the F-actin inhibits polymerization and annealing. Polymerization can be initiated not only by nucleation but also by uncapping at the barbed end (see Section 5) and by severing that creates free ends. Depolymerization is promoted by the dephosphorylation of ATP to ADP and the following release of γ-phosphate from the F-actin, which stimulates the dissociation of ADP-actin. Severing is also inseparably related to depolymerization. For example, the actin-depolymerizing factor (ADF)/cofilin induces depolymerization by enhancing either the severing or both the severing and the sequestering activities [23]. Enzymes that post-translationally modify actin can also regulate the processes of polymerization and depolymerization [24,25]. The Nebulin [26] and Drebrin-like protein DBN-1 [27] is able to stabilize F-actin in the sarcomeres of muscle cells. Tropomyosin protects F-actin from severing by cofilin and anneals gelsolin-severed actin fragments [28]. Abl2/Arg also stabilizes F-actin by slowing the process of depolymerization [29]. Cortactin stabilizes the branching mediated by Arp2/3 complex [30]. Actin cross-linking proteins possess multiple ABDs, thereby connecting to F-actin to form bundles (α-actinin, fimbrin, plastin, fimbacin) and networks (filamin) because of their unique geometry [31]. Branching is mediated by Arp2/3 complex and filamin [31,32]. Anchoring proteins such as ezrin-radixin-moesin (ERM) family proteins, talin, and spectrin connect F-actin to membrane proteins or lipids [33,34,35]. Finally, the myosin motor generates 2–5 pN of force to slide F-actin and transport cargo along the actin cable [36,37,38].

Further characterizations of ABPs in the past 40 years have revealed their multiple facets with overlapping functions. For example, some actin cross-linking proteins can also stabilize actin filaments by inhibiting depolymerization [39]. When the actin nucleating protein Arp2/3 complex generates branching and the branching dissociates, the Arp2/3 complex remains at the pointed end of the debranched filament and acts as a capping protein [2]. The monomer-binding protein profilin not only inhibits nucleation and elongation at pointed ends but also promotes elongation at the barbed end [40]. Recent research demonstrated that profilin also interacts with other ABPs and even with microtubules [41]. Finally, almost all AAPs also act as scaffolding proteins in order to interact with many other molecules. For example, filamins interact with over 150 binding partners [42,43]. Known binding partners can be found in the UniProt link. These examples are just the tip of the iceberg and are expected to grow in number.

## 4. How ABPs Interact with Actin

Biochemical analysis has found many ABDs and the structural analysis of AAP-actin complex has revealed a wide range of binding modes (Figure 3). This review catalogues them as follows:

Myosin motor domain (myosin) [44,45,46,47]: A structural analysis revealed a major difference between the cardiac and the other myosin isoforms in loops 3 and 2. Unlike skeletal and cardiac myosins, non-muscle myosins have loop 3 forming either electrostatic or hydrophobic interactions with the actin subdomain −1 (Figure 4c). This additional interaction may reduce the sliding velocity of non-muscle myosins [47]. Interestingly, the binding of myosin alters the structure of F-actin primarily at the DNase-binding loop (39-55aa) (Figure 3h), although such changes are dependent on isoforms [48]. A high-resolution structure of native vertebrate skeletal sarcomeres at different bands revealed by electron cryo-tomography shows that the two heads of double-headed myosin can interact with either a single actin filament or two separate actin filaments [49].

CH domain (filamin, spectrin beta chain, actinin, utrophin, plastin) [50,51,52,53,54]: High affinity actin-binding proteins such as filamin and actinin have two tandem CH domains. Although a single CH domain is not sufficient to interact with actin in some cases, some AAPs with a single CH domain (e.g., EH domain-binding protein 1) bind to actin [55,56].

Gelsolin domain (gelsolin, adseverin, villin, advillin, supervillin) [18]: The gelsolin domain comprises ~100 residues folded into five or six stranded β-sheets sandwiched between a long helix. Calcium binding induces the conformational changes of gelsolin to sever F-actin.

Cofilin: The binding of cofilin to actin leads to the conformational changes of actin, which are distinct from those of G-actin and F-actin [57]. Such conformational changes of actin result in the severing of F-actin.

Profilin: Profilin has two opposite regulatory functions in actin polymerization. When profilin binds to G-actin, it lowers the critical concentration for actin polymerization at the barbed end to promote assembly. The binding of proline-rich motif of Ena/VASP proteins, formins, and WASP/WAVE also support polymerization. However, when the barbed end is capped, profilin promotes depolymerization [58,59].

Actin-related protein (Arp): Arp2 and Arp3 are part of the Arp2/3 complex, which does not nucleate actin de novo, but promotes polymerization in the presence of both preexisting F-actin and a nucleation promotion factor (NPF) such as WASP/WAVE. Activated NPFs bind to both G-actin and the Arp2/3 complex to initiate the binding of Arp2/3 to the preexisting F-actin and nucleation [60,61].

Formin homology 2 (FH2) domain: Formin proteins stabilize unstable actin dimers and trimers by folding them using their dimeric FH2 domains, thus establishing stable actin filament nuclei [62,63].

β-Thymosin/WH2 domain: This is an intrinsically disordered actin-binding domain and is widely distributed, being predicted in more than 1800 proteins. The salt bridge between the highly conserved Glu334 of actin subdomain 3 and the LKKT/V motif of β-Thymosin/WH2 exerts a sequestering function [17,64].

Capping protein: Capping protein binds to the barbed end of F-actin to terminate assembly and can also prevent the association of NPFs to the barbed ends to indirectly stimulate the activity of Arp2/3 [65].

Twinfilin: Together with capping protein, twinfilin binds to two G-actins in an orientation that resembles the barbed end of F-actin, thereby accelerating polymerization. Twinfilin can also perturb the actin-capping protein interactions to dissociate the capping protein [66].

Alpha E catenin binds to F-actin in a force-dependent (catch bond) and an applied force direction-sensitive manner [67].

Leiomodin and tropomodulin: Despite their homology, tropomodulin caps the pointed end of the actin filament, whereas leiomodin nucleates actin and binds along the thin filaments in sarcomere [68,69].

Vinculin: Upon binding to F-actin, the vinculin tail domain undergoes unfolding, which is required for the bundling of F-actin [70].

MRTF-A PREL motif: The binding of G-actin to MRTF-A in the cytosol blocks the nuclear import of MRTF-A. The interaction is mediated by three actin-binding RPEL motifs in the regulatory domain of MRTF-A [71].

Cyclase-associated protein CAP1 and cofilin: The dimeric actin-binding domains of CAP bind two ADP-actin monomers to recharge ADP-G-actin with ATP [72].

Vitamin D binding protein: Although Vitamin D binding protein binds to the pointed end of actin filaments to terminate polymerization, it appears not to block the interaction with other actin molecules at the pointed end [73].

DNGR1: This is a C-type lectin receptor that binds to F-actin with weak affinity, despite having contact with three actin subunits. However, an avidity increase mediated by ligand binding can augment the binding strength [74].

Although we only listed actin-AAP complexes that currently exist in the protein data bank, computational analysis has also been used to build a model for some ABPs [75]. Nevertheless, many known AAPs possess uncharacterized ABD (smoothelin, fimbacin, etc.) [76,77] (Figure 3 and Appendix A). The structural geometry of AAPs influences their binding strength to actin filaments. For example, dimeric filamin A molecule has a much higher affinity to an actin filament than to a single subunit of the molecule. The specific L-shaped configuration of the molecule creates orthogonal actin networks, whereas dimeric alpha-actin forms bundles [31].

**Figure 3 ijms-23-02118-f003:**
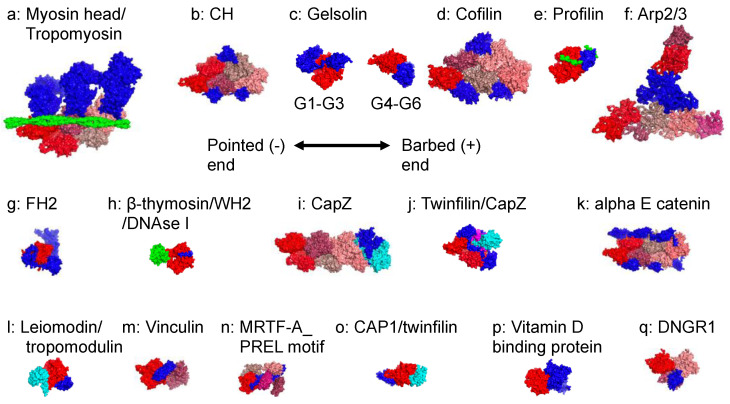
Surface models of the actin–ABP complexes: (**a**) Myosin motor domain (Cardiac myosin, 7JH7, tropomyosin is shown in green [47]). (**b**) CH domain (Utrophin, CH1, 6M5G [54]). (**c**) Gelsolin domain (gelsolin G1_G3, 1RGI [78], G4_G6, 1H1V, [79]). (**d**) Cofilin (5YU8 [57]). (**e**) Profilin (profilin, 2PBD [58], VASP is shown in green). (**f**) Actin-related protein (Arp2/3/actin, 7AQK, [60]). (**g**) Formin homology 2 (FH2) domain (Yeast Bni1p/actin, 1Y64 [62]). (**h**) β-Thymosin/WH2 domain (Wiskott–Aldrich syndrome protein (WASP), 2A3Z [64], DNAse I is shown in green). (**i**) Capping protein (CapZ, 7PDZ [65], CapZ alpha-1 is shown in cyan; CapZ beta is shown in blue). (**j**) Twinfilin and CapZ (7CCC [66], CapZ alpha-1 is shown in cyan; CapZ beta is shown in magenta). (**k**) Alpha E catenin (6WVT [67]). (**l**) Leiomodin and tropomodulin (4Z94 [68], leiomodin is shown in blue; tropomodulin is shown in cyan). (**m**) Vinculin (3JBI [70]). (**n**) MRTF-A PREL motif (2YJF [71]). (**o**) Cyclase-associated protein CAP1 and cofilin (6RSW [72], CAP1 is indicated in blue; twinfilin is indicated in cyan). (**p**) Vitamin D binding protein (1MA9 [73]). (**q**) DNGR1 (3J82 [74]). Actin in F-actin is shown from pointed (left) to barbed (right) end with red, tv red, raspberry, dark salmon, salmon, deep salmon, warm pink, and firebrick. ABPs are shown in blue. DNAse I is shown in green.

Only less than 40 ABPs have been structurally analyzed in a complex with G-actin or F-actin so far, and further analysis is necessary to reveal the molecular mechanisms of their functions and binding motifs.

## 5. Regulation of the Actin-AAP Interaction

The post-translational modification of actin regulates actin polymerization, stability, and interactions with AAPs. Surprisingly, 94 different side chains of actin have been found to be post-translationally modified including acetylation, arginylation, phosphorylation, and more, although not all modifications occur at the same time on the same molecule [24]. Interestingly, these modifications can modulate cell behaviors. For example, acetylation and arginylation are involved in filopodia and lamella formation, which ultimately regulate cell migration [80,81]. S-nitrosylation on Cys374 impairs binding to profilin-1 and reduces actin polymerization to regulate T-cell activation [82]. The phosphorylation of Tyr53 destabilizes F-actin to control the dendritic spine maturation and maintenance of long-term potentiation [83].

Post-translational modifications of AAPs also regulate their functions by modulating their interactions with actin. Since it is too overwhelming to describe all the modifications of AAPs in this review, we suggest readers refer to the review articles listed in Appendix A. For example, the phosphorylation of cofilin Ser3 diminishes its actin-binding activity and dephosphorylation enables its actin severing and depolymerizing activity and drives directional cell motility [84,85]. The binding of signaling molecules regulates the functions of AAPs as well. For example, polyphosphoinositides bind gelsolin and CapZ to uncap the barbed end of F-actin in order to induce actin polymerization [86]. Polyphosphoinositides also bind monomer-binding proteins such as profilin to separate G-actin in order to induce nucleation [87]. Calcium binding to gelsolin activates its severing activity [88]. CARMIL protein can inhibit capping protein in order to promote polymerization [89]. Recently, mechanical force has been recognized as a factor that regulates the actin cytoskeleton as well (Section 6).

## 6. How Mechanical Forces Influence the Actin Cytoskeleton

Mechanotransduction, a conversion of mechanical force into biochemical signal, occurs throughout life and the actin cytoskeleton is believed to mediate such biological signaling. For example, since the actin cytoskeleton is physically connected to the extracellular matrix through integrins and other transmembrane proteins, it follows that external mechanical forces such as pressure, shear force, and stretching would be transmitted to the actin networks. Internal mechanical forces generated by myosin contraction would also influence the conformation of the actin networks [90,91,92,93,94]. However, the need to reconstitute internal or external mechanical stress in vitro while quantifying protein–protein interactions remains a major obstacle to progress in this field of research. Once cells are lysed, force will be immediately lost, and it is not easy to manipulate molecules with physiologically relevant forces. Nevertheless, some AAPs have been demonstrated to be regulated by mechanical forces. For example, alpha-catenin binds vinculin in a force-dependent manner [95] and physiological forces can induce the conformational changes of talin to expose cryptic binding sites for vinculin [96]. Filamin can also be activated to interact with multiple molecules such as integrin, FilGAP, smoothelin, and fimbacin (LUZP1) upon mechanical stimulation [76,77,97]. Interestingly, while tension on actin filaments has no effect on cofilin binding and weakly enhances severing, ADF/cofilin can generate torsional stress on actin filaments to promote severing [98].

## 7. Actin and ABP as a Drug Target

Since actin and ABPs are involved in many human diseases because of their mutations or the anomaly of their upstream or downstream signaling pathways (Appendix A), drug development has been conducted targeting actin and AAPs. Appendix A summarizes a list of small molecules and antibodies (and nanobodies) that directly alter the functions of actin and AAPs. Although many small molecules have been developed to target molecules that directly or indirectly modulate AAPs, such drugs are not listed in the table.

Actin-binding drugs directly act on G-actin or F-actin. For example, phalloidin and jasplakinolide stabilize F-actin by binding at the interface of three actin subunits (Figure 4), although their effects on the kinetics of actin polymerization are different [99].

**Figure 4 ijms-23-02118-f004:**
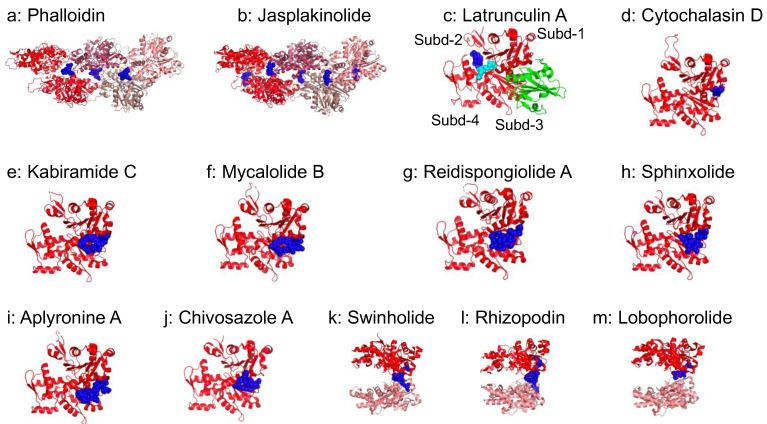
Actin-binding small molecules: (**a**) Phalloidin (7BTI [54]). (**b**) Jasplakinolide (6T23 [99]). (**c**) Latrunculin A, (1ESV [100], Gelsolin domain 1 is shown in green); subdomains of actin are indicated as Subd-1~4. ATP is shown in cyan. (**d**) Cytochalasin D (3EKS [101]). (**e**) Kabiramide C (1QZ5 [102]). (**f**) Mycalolide B (6MGO [103]). (**g**) Reidispongiolide A (2ASM [104]). (**h**) Sphinxolide (2ASO ([104]). (**i**) Aplyronine A (1WUA [105]). (**j**) Chivosazole A (6QRI [106]). (**k**) Swinholide (1YXQ [107]). (**l**) Rhizopodin (2VYP [108]). (**m**) Lobophorolide (3M6G [109]). Actin-binding small molecules are shown in blue. Actin shown from pointed (left) to barbed (right) end with red.

Latrunculin A binds near the actin nucleotide-binding site in a cleft between subdomains 2 and 4 and stabilizes actin monomers (Figure 4), thereby preventing them from repolymerizing into filaments [100]. Latrunculin also severs filaments and increases depolymerization at both ends of filaments [110]. Cytochalasin D instead binds to a hydrophobic cleft between actin subdomains 1 and 3, where many AAPs and also drugs target (Figure 4). Unlike latrunculin A, the mode of action of cytochalasin is complicated [101]. Cytochalasin D binds to the barbed end of F-actin with high affinity (Kd ≈ 2 nM) to inhibit the association and dissociation of actin monomers at the barbed end. Cytochalasin D also binds to G-actin with much lower affinity (Kd ≈ 2–20 μM, depending on the divalent cations). Moreover, cytochalasin D can induce actin dimer formation in the presence of magnesium and eliminate the lag phase in polymerization. ATP hydrolysis is also hastened by Cytochalasin D [111].

Kabiramide C and Mycalolide B belong to the trisoxazole family that mimics the binding of actin-capping proteins to actin in order to sever F-actin and cap the barbed end [112,113] (Figure 4). The macrolide derivative blocks cancer cell motility and invasion [103]. The reidispongiolide/sphinxolide family also binds to a hydrophobic cleft between actin subdomains 1 and 3 to sever F-actin and cap the barbed end [113]. Aplyronine A also binds to the same hydrophobic cleft of G-actin at a 1:1 ratio to depolymerize F-actin and inhibit polymerization. In addition, it possesses a potent antitumor effect [105]. Chivosazole A binds to G-actin to inhibit nucleation and polymerization and to sever F-actin. Interestingly, Chivosazole A selectively modulates the binding of ABPs to actin [106]. Swinholide A and rhizopodin are dimeric macrolides that stabilize the actin dimer at 1:2 stoichiometry to disrupt the actin cytoskeleton [107]. Lobophorolide, however, stabilizes an actin dimer at 2:2 stoichiometry [109]. Although many actin-targeting molecules have been developed and tested for the inhibition of the metastasis of cancer cell lines, none of these molecules were clinically used for their originally designed purpose, presumably because actin is used in normal physiology as well [114]. Nevertheless, these molecules are important research tools for studying the actin cytoskeleton, and some of them are used for other purposes. For example, raltergravir has been used to treat HIV/acquired immunodeficiency syndrome [115,116], and androgen receptors are the main target of flutamide [117].

Many drugs targeting ABPs that regulate muscle contraction and cell motility have been developed, and some of them are in clinical trial (Appendix A). Appendix A includes a link to ClinicalTrials.gov (https://clinicaltrials.gov/ct2/home, accessed on 9 February 2022), which is a database of clinical studies conducted around the world. For example, recently published phase III reports demonstrate the efficacy and safety of MYK-461 (Mavacamten) in targeting cardiac myosin for the treatment of hypertrophic cardiomyopathy [118,119,120]. Levosimendan targets troponin C with positive inotropic and vasodilatory effects and is being tested in phase III trials [121,122]. The first-in-class fascin inhibitor NP-G2-044 recently passed a phase Ia clinical trial to evaluate safety and anti-tumor activity in patients with advanced and metastatic solid tumors (https://clinicaltrials.gov/ct2/show/NCT03199586, accessed on 9 February 2022). Golodirsen, antisense oligonucleotide Vyondys 53, and related oligonucleotides are unique drugs that target the dystrophin gene to treat Duchenne muscular dystrophy (DMD) [123,124,125]. The binding of the oligonucleotide to DNA alters splicing to generate a functional gene product. FLNA-targeting drug PTI-125/Simufilam has recently been shown to reduce the cerebrospinal fluid biomarkers of Alzheimer’s disease pathology in a phase IIa study [126]. However, most of the AAP-targeting drugs listed in Appendix A failed in clinical trial or are not even in clinical trial, presumably because of their cytotoxicity to normal cells. However, some of the drugs that indirectly target AAPs appear to be promising for clinical use [123,127,128,129].

## 8. Conclusions and Future Perspective

We have listed 403 AAPs whose genes are independently coded on vertebrate DNA without including splice variants. In this review, we did not critically evaluate each AAP molecule, including current controversies, because many such review articles are available, as cited in Appendix A. Rather, we focused on comprehensively categorizing the known AAPs based on their functions and discussed the limitations of the current research concerning AAPs and the small molecules targeting actin and AAPs. Although some of the AAPs were identified from non-vertebrate organisms such as yeast and Dictyostelium discoideum, for which vertebrates have their homologues, such AAPs are not included in Appendix A. Future research is necessary to confirm the direct interaction of their vertebrate homologues with actin and to characterize more details of the reported interaction. For example, the actin interaction domains of some of the AAPs are not well-defined, and it is possible that only certain splice variants interact with actin. The identification and characterization of AAPs still requires laborious biochemical work in a wet laboratory. Current computational analysis to discover a new AAP using an AAP-binding motif, due to the diversity of binding modes of AAPs to G-actin and F-actin, is limited. In addition, not all the protein family members of known AAPs interact with actin. For example, although eight troponin genes have been identified in humans, and some of the gene products directly interact with actin, it is not clear whether other troponin family members also interact with actin. Despite the challenges, more AAPs will be discovered in the future, especially enzymes that post-translationally modify actin. Structural analysis is not an easy task, especially when an AAP interacts with filamentous actins with weak affinity in a state of multivalency. Since actin filaments and AAPs can be subjected to mechanical stress by external and myosin-mediated internal forces, it is possible for a cryptic binding domain to be exposed by mechanical stress. Such a domain has never been considered as a drug target. Although not many AAP-targeting drugs are promising for clinical use, AAPs have the potential to be used as biomarkers for some diseases; for example, some AAPs are upregulated in certain cancers [130,131,132,133,134].

## Figures and Tables

**Figure 1 ijms-23-02118-f001:**
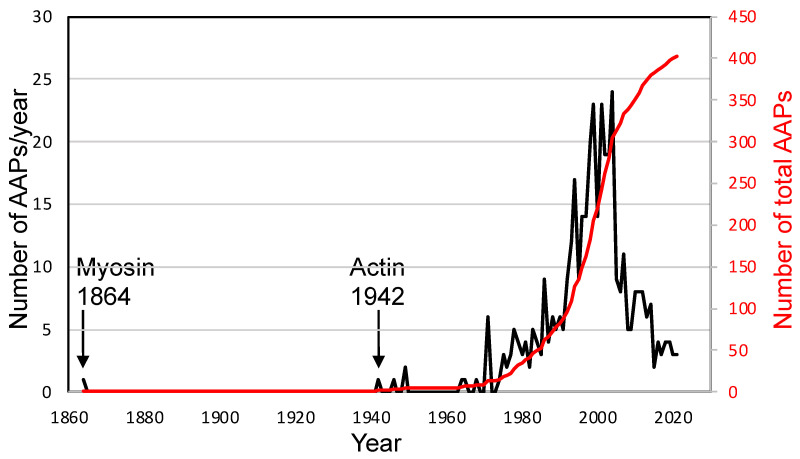
The number of actin-associated proteins (AAPs) published by year.

**Figure 2 ijms-23-02118-f002:**
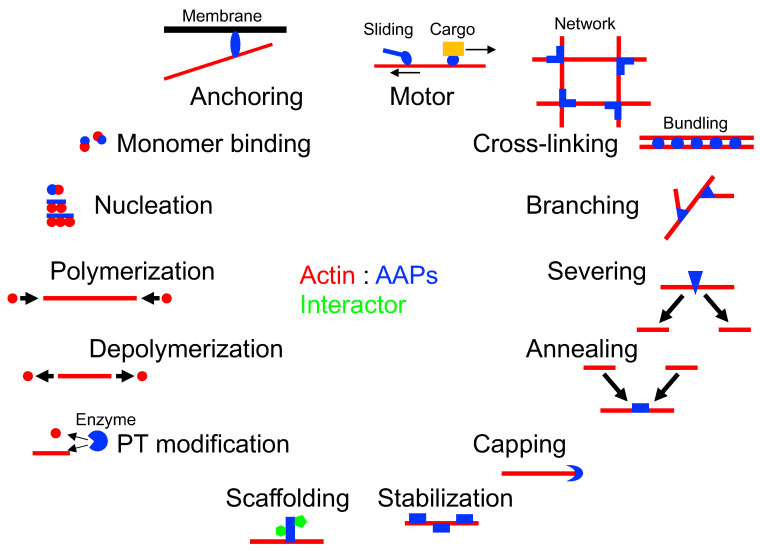
How AAPs interact with actin, regulate the assembly and disassembly of actin, connect, and move; post-translational (PT) modification. See also Appendix A.

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
