# Peer review of "Actin-Associated Proteins and Small Molecules Targeting the Actin Cytoskeleton"

_ijms, 2022, doi:10.3390/ijms23042118_

Round 1
Reviewer 1 Report
Gao and Nakamura reviewed the role of actin associated protein (AAP) for actin dynamics as well as compounds blocking actin dynamics. For this purpose, the authors explained the mechanism of actin polymerization and depolymerization and the regulatory role the AAPs have in this process. They classified the AAPs and highlighted some examples of each class, including the 3-D structures of the actin binding domains. Moreover, they summarized the compounds known to bind to actin, F-actin or to AAPs.
This article really enriches the community of cytoskeleton researchers. It is well-structured, the mechanism are competently explained and Table1 and Table 2 seem to completely list all known AAPs and their inhibiting compounds. It should be additionally mentioned that the fascin inhibitor NP-G2-044 passed the first clinical trial: “Phase 1A clinical trial of the first-in-class fascin inhibitor NP-G2-044 evaluating safety and anti-tumor activity in patients with advanced and metastatic solid tumors”. Furthermore, a critical discussion of the small molecules binding to actin is necessary, as for instance Flutamide mainly targets androgen receptors and Ralegravir is an integrase inhibitor used for HIV treatment. This non-actin targeting is true for further drugs listed in Table2 and should be labeled as such.
Author Response
We highly appreciate your careful reading and valuable suggestions which we incorporated to the manuscript as follows.
It should be additionally mentioned that the fascin inhibitor NP-G2-044 passed the first clinical trial: “Phase 1A clinical trial of the first-in-class fascin inhibitor NP-G2-044 evaluating safety and anti-tumor activity in patients with advanced and metastatic solid tumors”.
Added in page 9.
Inspired by the reviewer’s suggestion, we added a link to ClinicalTrials.gov (https://clinicaltrials.gov/ct2/home) in table 2.
Furthermore, a critical discussion of the small molecules binding to actin is necessary, as for instance Flutamide mainly targets androgen receptors and Ralegravir is an integrase inhibitor used for HIV treatment. This non-actin targeting is true for further drugs listed in Table2 and should be labeled as such.
When non-actin targeting is known, it is indicated in “Target disease and function” and a link to Clinical Trial in table 2. It is also mentioned on page 9.
Reviewer 2 Report
This manuscript aims to review both actin associated proteins and actin binding proteins which are listed in Table 1. A further listing of actin targeting compounds is presented in Table 2. These comprehensive listings are potentially useful although there is limited novelty in these lists. In addition, there is little critical evaluation of the entries in these lists. Figure 3 provides coverage of the surface interaction between actin and a number of these actin interacting proteins, but the detail provided in the figure is insufficient to draw any conclusions regarding key features of the interactions. This paper is largely a compendium of literature reports with no new insights. It is surprising that the work of Stefan Raunser on the structure of the myosin/actin/tropomyosin interface which provides the greatest molecular detail on the interaction of actin interacting proteins with actin is nor covered. In the absence of any critical evaluation, it is difficult to support publication of this compendium in its current form.
Author Response
Although we admit that we did not discuss much about the surface interaction between actin and the actin interacting proteins, such details are not a core of our manuscript because many review papers regarding structural basis of the interactions are available. Rather we focused on comprehensively categorizing known actin-associated proteins (AAPs) based on their functions and discussed limitations of the current research on AAPs and small molecules targeting actin and AAPs as mentioned in the conclusion. We believe that no review article exists that comprehensively classified AAPs as shown in Figure 1 (previous review articles generally focused on actin-binding proteins, hence a part of our proposed classification) and listed all known AAPs. We also believe that we provide insight to the reader by examining various key challenges apparent from the extant research, and suggesting new research opportunities based on a thorough review of past work. Stefan Raunser’s work has been cited on page 4 as suggested.
Reviewer 3 Report
In table 2 the effective concentrations for living cells and for cell-free applications should be distinguished. For experimental work, information would be appreciated whether a substance is membrane permeable, implying that it acts on living cells.
A list of contents of the substances in alphabetical order would be helpful.
Latrunculin: Does it really depolymerize, or is it the cellular turnover that results in depolymerization when polymerization is blocked?
Minor points:
In the tables, division of the words is often incorrect, e. g. embryoni – c. There are some spelling errors.
Reference 48 in table 2: Knolker should be Knölker (Knoelker).
“Nanobodies” as L-plastin inhibitors should be specified.
Author Response
Thank you for your careful reading and valuable suggestions which we incorporated to the manuscript as much as possible as follows.
In table 2 the effective concentrations for living cells and for cell-free applications should be distinguished.
The effective concentrations for living cells and organism are mentioned in the table. Otherwise, effective concentrations reported are for cell-free applications. This is indicated at the bottom of the table.
For experimental work, information would be appreciated whether a substance is membrane permeable, implying that it acts on living cells.
If a substance is used for living cells, it is likely membrane permeable unless it is incorporated by different mechanism such as endocytosis. Such information is limited in the literature.
A list of contents of the substances in alphabetical order would be helpful.
The list can be sorted in alphabetically order on excel using the excel file we provided. However, we would prefer the current order because the substances are grouped by their targets.
Latrunculin: Does it really depolymerize, or is it the cellular turnover that results in depolymerization when polymerization is blocked?
Fujiwara et al demonstrated that Latrunculin not only blocks polymerization by sequestering actin monomers but also severs filaments and increases the depolymerization rate (Curr Biol. 2018 Oct 8;28(19):3183-3192.e2. Latrunculin A Accelerates Actin Filament Depolymerization in Addition to Sequestering Actin Monomers). This literature has been added in the table and mentioned in the text (page 8).
Minor points:
In the tables, division of the words is often incorrect, e. g. embryoni – c. There are some spelling errors.
This is due to space limitation of the word table. Character size was reduced to avoid this problem.
Reference 48 in table 2: Knolker should be Knölker (Knoelker).
Amended.
“Nanobodies” as L-plastin inhibitors should be specified.
Specified.
Round 2
Reviewer 3 Report
I have an additional question to the authors. Did they purposely omit the paper by Kepiro et al. in Angew. Chem. Int. Ed. 53, 8211-8225 (2014) on the non-toxic and photostable para-Nitroblebbistatin?
I do not understand the altered sentence on lines 159-160 of the main text.
Author Response
I have an additional question to the authors. Did they purposely omit the paper by Kepiro et al. in Angew. Chem. Int. Ed. 53, 8211-8225 (2014) on the non-toxic and photostable para-Nitroblebbistatin?
Thank you for your suggestion. The following two literatures have been added in table 2.
- Kepiro M, Varkuti BH, Vegner L, Voros G, Hegyi G, Varga M, et al. para-Nitroblebbistatin, the non-cytotoxic and photostable myosin II inhibitor. Angew Chem Int Ed Engl 2014, 53(31): 8211-8215.
- Varkuti BH, Kepiro M, Horvath IA, Vegner L, Rati S, Zsigmond A, et al. A highly soluble, non-phototoxic, non-fluorescent blebbistatin derivative. Sci Rep 2016, 6: 26141.
I do not understand the altered sentence on lines 159-160 of the main text.
“double-head myosin interacts with not only the same actin filaments but also splits between two actin filaments”
We have rephrased as follows.
the two heads of double headed myosin can interact with either a single actin filament or two separate actin filaments